# Transformation of aqueous protein attenuated total reflectance infra-red absorbance spectroscopy to transmission

**Key words:**
Attenuated total reflectance; infrared spectroscopy; protein; structure; transmission; water

**Author for correspondence:**
Alison Rodger,
E-mail: Alison.Rodger@mq.edu.au

Alison Rodger[1] , Michael J. Steel[2] , Sophia C. Goodchild[1] ,

Nikola P. Chmel[3] and Andrew Reason[4]

[1]Department of Molecular Sciences, Macquarie University, Sydney, NSW 2109, Australia; [2]Department of Physics and Astronomy, Macquarie University, Sydney, NSW 2109, Australia; [3]Department of Chemistry, University of Warwick, Coventry CV4 7AL, UK and [4]BioPharmaSpec Ltd., Lido Medical Centre, St. Saviour, Jersey JE2 7LA, UK

## Abstract

Infrared (IR) spectroscopy is increasingly being used to probe the secondary structure of proteins, especially for high-concentration samples and biopharmaceuticals in complex formulation vehicles. However, the small path lengths required for aqueous protein transmission experiments, due to high water absorbance in the amide I region of the spectrum, means that the path length is not accurately known, so only the shape of the band is ever considered. This throws away a dimension of information. Attenuated total reflectance (ATR) IR spectroscopy is much easier to implement than transmission IR spectroscopy and, for a given instrument and sample, gives reproducible spectra. However, the ATR-absorbance spectrum varies with sample concentration and instrument configuration, and its wavenumber dependence differs significantly from that observed in transmission spectroscopy. In this paper, we determine, for the first time, how to transform water and aqueous protein ATR spectra into the corresponding transmission spectra with appropriate spectral shapes and intensities. The approach is illustrated by application to water, concanavalin A, haemoglobin and lysozyme. The transformation is only as good as the available water refractive index data. A hybrid of literature data provides the best results. The transformation also allows the angle of incidence of an ATR crystal to be determined. This opens the way to using both spectral shape and spectra intensity for protein structure fitting.

## Introduction

Protein infrared (IR) data, especially from the amide I (1,700–1,600 cm$^{-1}$) band, are recognised as containing information about protein secondary structure. Normalised and second derivative spectra have both been used to estimate protein secondary structures (Chittur, 1998*a*, 1998*b*). The magnitude of the water signal (21.8 mol$^{-1}$ dm$^3$ cm$^{-1}$ at 1,643 cm$^{-1}$) means a path length of less than 8 μm is required (Bertie and Eysel, 1985; Bertie *et al.*, 1989; Venyaminov and Prendergast, 1997; Max and Chapados, 2009), and the protein in a 20 mg ml$^{-1}$ aqueous sample contributes about 2% of the signal. Although the Beer–Lambert law for absorbance, *A*,

$$A = \log_{10}\left(\frac{\mathrm{I}_0}{\mathrm{I}_{\mathrm{final}}}\right) = \varepsilon C \ell, \tag{1}$$

(where $\mathrm{I}_0$ is intensity incident on the sample, $\mathrm{I}_{\mathrm{final}}$ the final intensity, $\varepsilon$ is the wavelength-dependent extinction coefficient, *C* is concentration, and $\ell$ is the total path length) holds for transmission IR spectroscopy, it is extremely challenging to collect sample and baseline spectra with the same, accurately known, path length on the micron scale. Protein IR spectroscopy has, therefore, often been performed in D$_2$O, whose corresponding band occurs at 1,209 cm$^{-1}$. However, biological samples contain H$_2$O, and biopharmaceutical products are formulated in H$_2$O. Raman spectroscopy might be an alternative as Raman is less influenced by H$_2$O scattering than IR absorbance is affected by H$_2$O absorbance. However, the side chain contributions to Raman spectra seem to be sufficient to invalidate attempts at secondary structure fitting from Raman spectra (Pinto-Corujo, 2020).

An alternative approach for IR spectroscopy was provided by the invention of attenuated total reflectance (ATR, Fig. 1) IR spectroscopy in 1959–1960 by Harrick and Fahrenfort (Harrick, 1960, 1965; Harrick and du Pré, 1966; Mirabella, 1992). With ATR, the sample is placed in contact with a transparent dense crystal whose refractive index (speed of light in a vacuum, $c_0$, divided by speed of light in the medium, *c*) is so much higher than that of the sample that if the angle of incidence (with 0° being perpendicular to the surface) is high enough, the light beam does not pass into the sample but is totally internally reflected (Fig. 1*a*). Maxwell's equations

(see Appendix) require that there is electric (and magnetic) field intensity from the light present above the surface – this is usually referred to as an evanescent *wave*, even though the light beam does not actually propagate away from the surface. These fields, which decay away from the surface, interact with the sample and can cause the molecule to move to an excited state so that less light reaches the detector than in the absence of the sample. Thus, a spectrum is generated that depends on the absorbance of the sample.

ATR spectroscopy has been widely adopted because of its ease of use. However, the spectrum that is measured with ATR is not the same as in transmission and differs from instrument to instrument: the relative magnitudes of different parts of the spectrum depend on the wavenumber, the angle of incidence of the light and the refractive indices of both the crystal and the sample – with the latter in turn being dependent on its absorbance. ATR-IR has, therefore, seldom been used quantitatively.

A cursory perusal of the literature leads one to believe the equations required to relate ATR-IR spectra to transmission spectra are to be found somewhere, as everyone quotes the same equations. However, we have failed to find coherent derivations, which hindered our ability to transform data quantitatively to an instrument-independent spectrum to be used for regulatory purposes. The goal of our work was therefore to establish a method to transform aqueous protein ATR spectra into what could be measured in a transmission experiment. In this way, we can benefit from the reproducible simplicity of the ATR experiment, without having instrument-to-instrument variations.

This paper is structured in terms of increasing complexity, beginning with how to collect data, then how to transform an ATR aqueous protein spectrum to transmission and finally, the theory behind the equations used in the transformation is derived in the Appendix. Understanding of the theory is not required to perform the transformations, but it helps to understand the literature. Most of the analysis in this paper relates to water, for which reasonably accurate transmission and refractive index data are available. Application to three aqueous protein examples are also included.

## Methods

### Materials

18.2 MΩ water was used for all experiments. Proteins were obtained from Sigma–Aldrich (Gillingham, UK).

### Instrument

IR data were collected with a Jasco V-470 IR spectrometer at 4 cm$^{-1}$ resolution with a CaF$_2$ demountable cell without spacer (approximately, 1 μm pathlength) or a PIKE MIRacle single reflection ZnSe ATR unit and a triglycine sulfate (TGS) detector. A Specac Golden Gate single reflection ATR unit with a Jasco FTIR-4200 IR spectrometer was also used for comparison (Appendix). We chose TGS as it has a wider dynamic range than mercury cadmium telluride (MCT) detectors, and we found it was more reproducible, as well as having the pragmatic advantage of not requiring liquid nitrogen. ZnSe has reasonable energy throughput (~25–30% of that in a transmission experiment) over the range we wanted and a refractive index of $n_i$ ~ 2.4. Although germanium ATR crystals have a higher refractive index than ZnSe, which is attractive for reducing the wavenumber dependence of the signal, its smaller penetration depth means any surface effects (such as protein binding or

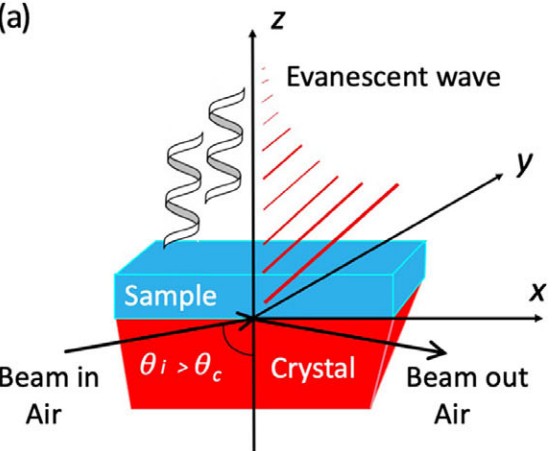

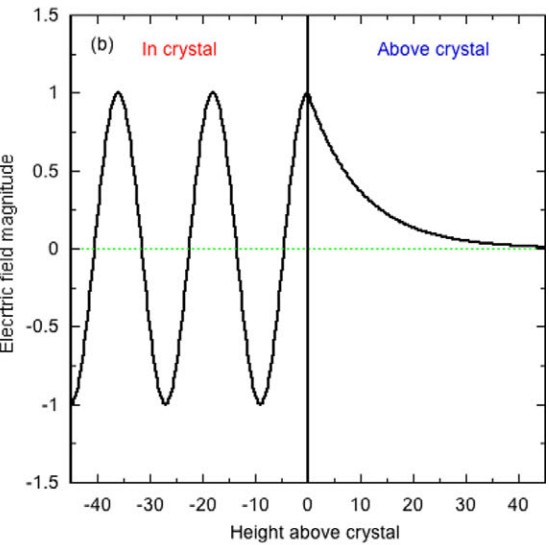

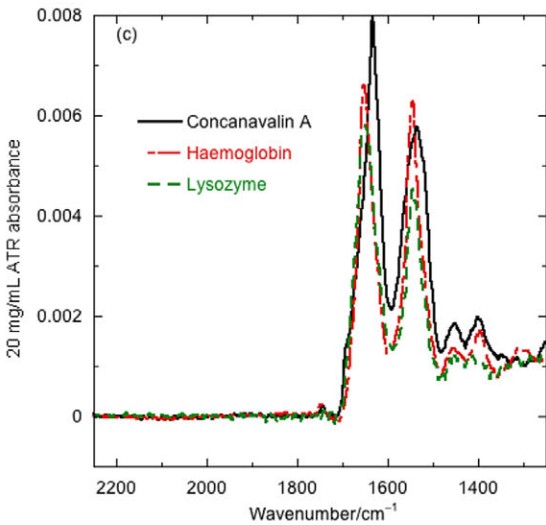

**Fig. 1.** (*a*) Configuration of an attenuated total reflectance infrared (ATR-IR) experiment. The crystal surface is the *x*–*y* plane; +*z* is above the crystal and the incident light beam propagates in the *x*–*z* plane. (*b*) Light amplitude above and below the crystal surface. (*c*) Overlay of ATR-IR spectra of 20.0 mg ml$^{-1}$ aqueous samples of three proteins after water baseline correction. Data collected with a Jasco V-470 IR spectrometer equipped with a PIKE MIRacle single reflection ZnSe ATR unit.

orientation at the surface) are more significant. As baseline-corrected aqueous protein ATR spectra in the amide I/amide II region are proportional to concentration up to at least 50 mg ml$^{-1}$ (data not shown) and since water essentially determines the penetration depth of the light beam (see below), we can conclude that concentration or significant orientation of the proteins is not occurring at the surface.

ATR-IR spectra were measured directly after taking a new background (air only) spectrum. Proteins were dissolved in water. A water baseline spectrum was subtracted from the aqueous protein spectrum. Baseline correction was sometimes improved by modifying the water spectrum by up to 1% of its magnitude. Baseline correction was deemed of good quality if the spectrum was flat in the liquid water libration/bend region (2,100 cm$^{-1}$). If necessary, we then added or subtracted a water vapour spectrum until the water vapour bands around 1,700 cm$^{-1}$ disappeared. The water vapour spectrum was generated from the difference between successive air spectra and scaled to the magnitude required. In general, if backgrounds (air only) were collected immediately prior to measuring a spectrum, this was not needed. Spectra were zeroed at 4,400 cm$^{-1}$ unless the spectrum had been collected over a narrower window in which case, for aqueous samples, we used an intensity at 4,000 cm$^{-1}$ of about 0.001 or at 2,620 cm$^{-1}$ of about 0.003 as 'zero'.

Protein concentrations used in this work are in mg ml$^{-1}$ of pure protein determined from measuring the $A_{280}$ of the solution in 1 mm pathlength cuvettes and using protein theoretical extinction coefficients determined from the sequence.

## Parameters used

Water transmission spectra were scaled to the literature extinction coefficient of 21.8 mol$^{-1}$ dm$^3$ cm$^{-1}$ at 1,643 cm$^{-1}$ (concentration 55.506 M). The initial refractive index of water was determined from Bertie *et al.* (1989) and Bertie and Eysel (1985) by linearly interpolating between published data points (referred to as Bertie). It was refined using Max and Chapados' (Max and Chapados, 2009) above ~3,350 cm$^{-1}$ (with manual smoothing in the cross-over region). The refractive index of ZnSe was taken from Madelung *et al.* (1999). The angle of incidence of our ATR PIKE unit was determined by establishing the value of $\theta_i$, which gave the best overlay of a transmission spectrum and a transformed-ATR spectrum. The value was 44.7°. The Specac unit (see Appendix), which had been aligned by eye, proved to have $\theta_i$ = 48.0°.

## Calculations

Excel was used to implement the transformations between transmission and ATR. The supplementary material contains a spreadsheet with typical calculations and data for the plots. Literature water absorbance and refractive index data were used (21.8 mol$^{-1}$ dm$^3$ cm$^{-1}$ at 1,643 cm$^{-1}$ and ~101 mol$^{-1}$ dm$^3$ cm$^{-1}$ at 3,408 cm$^{-1}$ (Bertie and Eysel, 1985; Bertie *et al.*, 1989; Venyaminov and Prendergast, 1997; Max and Chapados, 2009).

## Results

Fig. 1*c* shows baseline-corrected ATR-IR spectra for three proteins of different secondary composition (fraction helix: concanavalin = 0.04; lysozyme = 0.42; haemoglobin = 0.76) at 20.0 mg ml$^{-1}$ of protein in water. As the mean residue weights of the proteins are concanavalin: 108.5 Da, lysozyme: 111.6 Da and haemoglobin:

112.5 Da, the magnitude differences are more significant when molar concentrations are used. This means that discarding intensity information by normalising the amide I band is throwing away a dimension of information. With analogous circular dichroism data, normalisation leads to poor α-helix estimates (Hall *et al.*, 2014). We are therefore motivated to provide a quantitative transformation of ATR data to transmission.

### Transmission versus ATR: Step 1, the penetration depth

Aqueous protein absorbance spectra are dominated by water, so we first consider how to relate transmission and ATR water spectra. Fig. 2*a* shows the overlay of scaled transmission (dashed red line) and ATR (black line) water spectra. The 1,650 and 3,250 cm$^{-1}$ bands correspond to water vibrations. The 2,125 cm$^{-1}$ band is a combination liquid water bend + libration band (Max and Chapados, 2009), which is absent from water vapour (and proteins) and is thus extremely useful for ensuring good baseline subtraction – this region should be flat as in Fig. 1*c*. 2,350 cm$^{-1}$ is $CO_2$ and indicates the stability of purging. The presence of water vapour is indicated by oscillations from 1,700 cm$^{-1}$ downwards.

The relative magnitudes of the ATR and transmission spectra vary with wavenumber due in part to the 'depth of penetration' or effective path length, $d_p$, of an ATR experiment, which here ranges from 0.34 μm (at 4,000 cm$^{-1}$) to 1 μm (at 1,500 cm$^{-1}$) (see below and Fig. A1*a*). As shown in the Appendix,

$$d_p = \frac{1}{\alpha} = \frac{1}{k_i \sqrt{\sin^2\theta_i - n_t^2/n_i^2}} = \frac{\lambda_0}{2\pi n_i \sqrt{\sin^2\theta_i - n_t^2/n_i^2}}, \quad (2)$$

where $n = c_0/c$ is the refractive index, $i$ refers to incident light in the crystal, $t$ refers to light transmitted above the crystal surface, $\lambda_0$ is the free space wavelength and $\theta_i$ is the angle of incidence of the light on the crystal (Fig. 1*a*). The wavenumber variation in $d_p$ is often calculated assuming the refractive index, $n_t$, is independent of sample (in our case, water plus protein) absorbance. This correction (Fig. 2*a*, dashed green line) transforms the ATR spectrum to be similar in magnitude to the transmission. Further improvement comes from including the wavenumber dependence of the refractive index (Fig. 2*a*, blue dotted line). However, the difference is still larger than the protein absorbances of Fig. 1*c*, so the problem is not solved just by considering $d_p$.

### Transmission versus ATR: Step 2, the light intensity

In ATR-IR, the light beam interacts with the sample *via* the electric field that exists above the ATR crystal. The two variable aspects of the flux of light that analyte molecules encounter are:

 (i)  the intensity of the light just above the crystal surface and
(ii)  how quickly the intensity of the light decays away from the crystal surface

Both of these factors depend on the nature of the ATR crystal, the frequency of the light, the sample absorbance as a function of frequency and the parameters of the instrument being used. What the instrument nominally plots as absorbance in the ATR experiment is

$$\text{Nominal absorbance} = A_{ATR}$$
$$= \log_{10}\left(\frac{\text{light hitting the detector for the reference}}{\text{light hitting the detector for the sample}}\right) \quad (3)$$
$$= \log_{10}\left(\frac{I_{air}}{I_{sample}}\right),$$

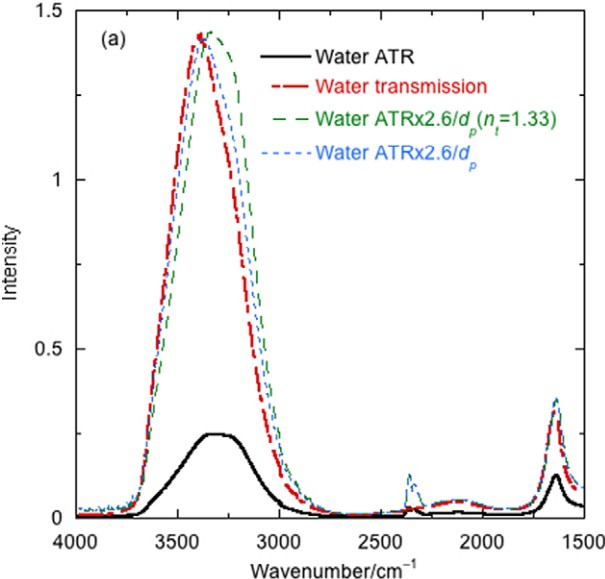

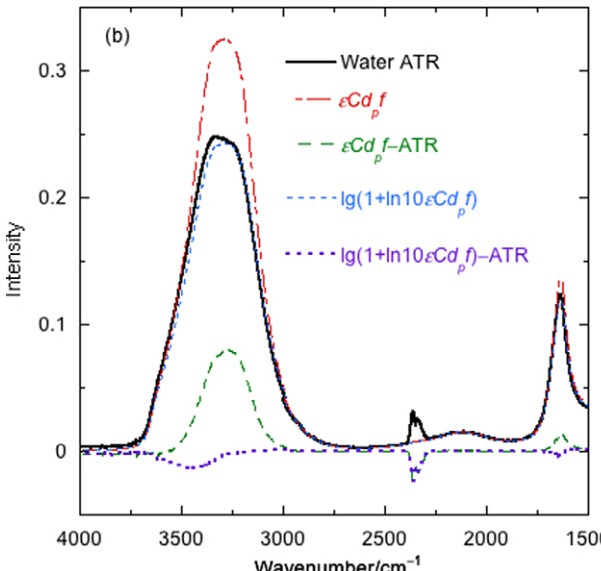

**Fig. 2.** Water infrared (IR) spectra. (*a*) Transforming ATR towards transmission with penetration depth corrections: Overlay of: ZnSe ATR water spectrum; transmission water spectrum in a cell without spacer, $\ell = 2.6\ \mu$m (based on absorbance at 1,643 cm$^{-1}$); conversion according first term in Eq. (11) using the infinite wavenumber fixed refractive index and conversion according first term in Eq. (11) using the wavelength-dependent refractive index (Fig. A1). (*b*) Transforming transmission spectra to ATR: Overlay of ZnSe ATR water spectrum with first-order transformation (Eq. (10), term 1) transformation and full transformation (Eq. (9)). $\theta = 44.7°$, $C_{water} = 55.506$ M, $\varepsilon_{1643} = 21.8$ mol$^{-1}$ dm$^3$ cm$^{-1}$, water refractive index data from Bertie *et al.* (1989) and Bertie and Eysel (1985) (Bertie). $d_p$ in cm is used in all calculations. lg denotes log$_{10}$. The difference between the ATR spectrum calculated from transmission data and the ATR spectrum is also shown.

where $\mathbb{I}_{air}$ is the intensity that reaches the detector when only air is above the ATR crystal (the background) and $\mathbb{I}_{sample}$ is the intensity that hits the detector in the presence of a sample on the crystal. The component strengths of the incident electric field light in the ATR experiment is different in the *x*, *y* and *z* directions of Fig. 1. They are also different just below and just above the surface of the crystal and different in the crystal from in air. As derived in the Appendix, the

light intensity components transmitted to just above the surface of the crystal are:

$$\begin{aligned}
\mathbb{I}_{t0x} &= \mathbb{I}_{air}\mathbb{f}_{AtoC}\mathbb{f}_x \\
\mathbb{I}_{t0y} &= \mathbb{I}_{air}\mathbb{f}_{AtoC}\mathbb{f}_y \\
\mathbb{I}_{t0z} &= \mathbb{I}_{air}\mathbb{f}_{AtoC}\mathbb{f}_z,
\end{aligned} \tag{4}$$

where, for example, $\mathbb{I}_{t0x}$ is the *x* component of the light intensity, $\mathbb{f}_{AtoC}$ is the air to crystal change in light intensity and $\mathbb{f}_x$ is the *x*-direction change in light intensity of an unpolarised beam on passing from below to above the crystal surface. It follows from Fresnel's equations (see Appendix) that

$$\begin{aligned}
\mathbb{f}_x &= \frac{2n_i^2 n_t \cos^4\theta_i \left(n_i^2 \sin^2\theta_i - n_t^2\right)}{\left(n_i^4 \sin^2\theta_i - n_i^2 n_t^2 + n_t^4 \cos^2\theta_i\right)} \\
\mathbb{f}_y &= \frac{2n_i^2 n_t \cos^2\theta_i}{\left(n_i^2 - n_t^2\right)} \\
\mathbb{f}_z &= \frac{2n_i^4 n_t \cos^2\theta_i \sin^4\theta_i}{\left(n_i^4 \sin^2\theta_i - n_i^2 n_t^2 + n_t^4 \cos^2\theta_i\right)}
\end{aligned} \tag{5}$$

$$\mathbb{f}_{AtoC} = \left(\frac{2}{1+n_i}\right)^2. \tag{6}$$

For an unpolarised beam and an unoriented sample, the total light intensity at the surface is simply

$$\mathbb{I}_0(z) = \left(\mathbb{I}_{t0x} + \mathbb{I}_{t0y} + \mathbb{I}_{t0z}\right). \tag{7}$$

Eqs. (4)–(7) are essential for quantitative inter-transformation between ATR and transmission absorbance intensities and differ from those found in the literature (Harrick, 1960, 1965; Harrick and du Pré, 1966) by

(i) $\mathbb{f}_{AtoC}$, which recognises the loss of light intensity in going from air into the crystal (which affects how much light the sample sees);
(ii) inclusion of the dependence on $\theta_i$ of the *x* and *z* component intensities just below the surface and
(iii) a factor of 2 included for an unpolarised light beam.

At any position above the surface, the intensity is attenuated by both the decay of the evanescent wave plus any absorbance, so (see Appendix)

$$\begin{aligned}
\mathbb{I}_t(z) &= \left(\mathbb{I}_{t0x} + \mathbb{I}_{t0y} + \mathbb{I}_{t0z}\right)e^{-\left(2/d_p + 2\varepsilon C \ln 10\right)z} \\
&= \mathbb{I}_{air}\mathbb{f}_{AtoC}\left(\mathbb{f}_x + \mathbb{f}_y + \mathbb{f}_z\right)e^{-\left(2/d_p + 2\varepsilon C \ln 10\right)z} \\
&= \mathbb{I}_{air}\mathbb{f}e^{-\left(2/d_p + 2\varepsilon C \ln 10\right)z},
\end{aligned} \tag{8}$$

where $\varepsilon C$ is the extinction coefficient (absorbing power) times concentration and $\mathbb{f}$ is as defined in Eq. (8).

In order to determine how the frequency dependence of the light intensity affects the nominal ATR absorbance signal (Eq. (3)), we integrate the absorbance from $z = 0$ to infinity. Thus (see Appendix),

$$A_{ATR} = \log_{10}\left(1 + \ln 10\, \varepsilon C d_p \mathbb{f}\right) \tag{9}$$

$$A_{ATR} = \varepsilon C d_p \mathbb{f} - \frac{\ln 10}{2}\left(\varepsilon C d_p \mathbb{f}\right)^2 + h.o.t. \tag{10}$$

(*h.o.t.* indicates higher order terms). Eq. (9) can be used as an accurate transformation of a transmission spectrum to ATR. Eq. (10) enables the reverse transformation. Its first term is comfortingly familiar as the transmission absorbance intensity times f, but it is only an approximation.

### Transforming transmission spectra to ATR spectra

Fig. 2*b* shows an overlay of the experimental water ATR spectrum and the water ATR spectrum calculated from transmission data using Eq. (9) (not an approximation), Bertie's water refractive index (not perfect, see Appendix) and an angle of incidence of 44.7°. The spectrum from only the first term in Eq. (10) is also shown. The Eq. (9) spectrum (dashed green line) overlays with the experimental ATR except in the regions of absorption maxima (see residual spectra). Literature water refractive indices vary noticeably (Fig. A1), especially in regions of high absorbance (Bertie and Eysel, 1985; Bertie *et al.*, 1989; Max and Chapados, 2009). If we choose to use Bertie's data until about 3,350 cm$^{-1}$ and Max and Chapados' after that point, we get better results for two different instruments and ATR units (Fig. A1). We therefore conclude that this hybrid refractive index is best for applications. The first term in Eq. (10) is reasonably good for absorbances less than 0.1 and corresponds to the low absorption limit of the literature (Harrick, 1965; Milosevic, 2004). However, for aqueous proteins, due to the water absorbance, the low absorption limit is not satisfactory. Including the second term improves the result as much as the quality of available refractive index data allows. If one normalises the data to 1 at the maximum near 1,645 cm$^{-1}$ (a common practice), the discrepancies are masked in the amide I region of the spectrum but are still there and affect any further applications of the data.

### Transforming ATR spectra to transmission spectra

As ATR spectra vary from instrument to instrument, we are more likely to want to transform ATR spectra into transmission than the converse. As shown in Fig. 2*b*, for water, we need more than the first term of Eq. (10), so we use the first two terms to give

$$\varepsilon C \approx \frac{1 \pm \sqrt{1 - 2A_{ATR}\ln 10}}{\ln 10\, d_p\, f} = \frac{A_{ATR}}{d_p\, f} + \frac{A_{ATR}^2 \ln 10}{2 d_p\, f}. \quad (11)$$

Multiplication by the experimental path length is required to turn Eq. (11) into transmission absorbance, and division by concentration gives extinction coefficients.

The difference between the solid black and dashed green lines in Fig. 3 at 3,500 cm$^{-1}$ is a combination of losing the higher order terms (a small effect, see above) and the experimental errors in water transmission absorbance and refractive index in that region. However, the intensities and shapes give us confidence that even in the high-absorbing regions of the spectrum, Eq. (11) is adequate.

### Application to aqueous protein spectra

To perform the ATR to transmission transformation for aqueous protein samples, we first assume $d_p$ is the same for both the protein in buffer and the buffer since amide I protein absorbance is about 2% of the $H_2O$ absorbance at 20 mg ml$^{-1}$ protein. Then, from Eq. (11), removing the *ATR* subscripts (so *A* denotes the nominal ATR absorbance), we write

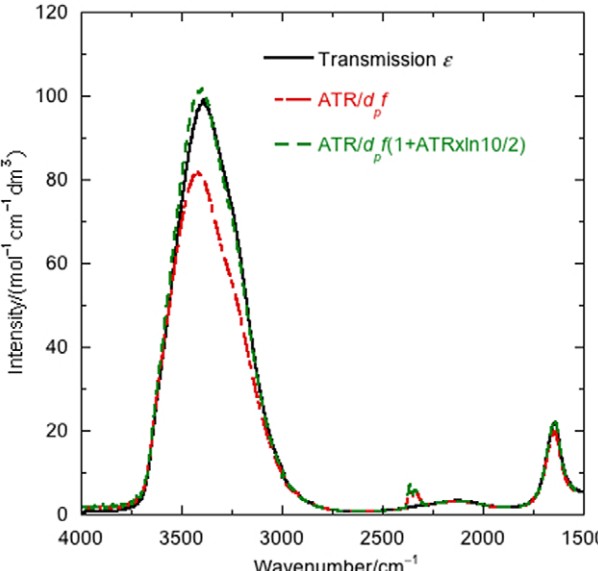

**Fig. 3.** Overlay of water extinction coefficients for transmission and first- and second-order terms in the transformation of ATR to transmission (Eq. (11)) determined using $\theta = 44.7°$, 55.506 M as concentration, 21.8 mol$^{-1}$ dm$^3$ cm$^{-1}$ as the extinction coefficient and Bertie's refractive index data (Bertie and Eysel, 1985; Bertie *et al.*, 1989). $d_p$ in cm is used in calculations.

$$
\begin{aligned}
(\varepsilon C)_P &= (\varepsilon C)_{PW} - (\varepsilon C)_W \\
&\approx \frac{A_{PW} - A_W + A_{PW}^2 \ln 10/2 - A_W^2 \ln 10/2}{d_p\, f} \\
&= \frac{A_P + (A_P + A_W)^2 \ln 10/2 - A_W^2 \ln 10/2}{d_p\, f} \quad (12) \\
&= \frac{A_P}{d_p\, f} + \frac{A_P(A_P + 2A_W)\ln 10}{2 d_p\, f},
\end{aligned}
$$

where *P* refers to the protein contribution to absorbance, *PW* denotes the absorbance of the protein plus water/buffer solution and *W* denotes the water/buffer absorbance. $A_P$ is determined using the baseline correction process outline in the Methods section. In what follows, we use Bertie's refractive index data modified by Max and Chapados' data at high wavenumbers (see Appendix).

When the different levels of correction are applied and the results all scaled to unit normalised absorbance (Fig. 4*a* for lysozyme) at the amide I maximum, we reduce the comparison to band shape only. Most protein IR spectroscopy is presented in this way to avoid needing accurate concentrations and path lengths. As illustrated for lysozyme in Fig. 4*a*, everything looks artificially good for the normalised amide I band, though the amide II is poor except for the second-order correction. Alternatively, one could make the amide II look good at the expense of the amide I band.

When we do not normalise the spectra, it is clear from Fig. 4*b* that the second-order correction is required. Concanavalin (Fig. 2*c*) and haemoglobin (Fig. 2*d*) present a similar picture to that of lysozyme.

The apparently disappointing aspect of Fig. 4*a,b* is the 1–2 cm$^{-1}$ mismatch of the transmission spectrum and the second-order ATR to transmission transformation (which includes the second term in Eq. (12) with full wavenumber dependence of the refractive index in $d_p$ and f ). Rather than optimising data collection and baseline

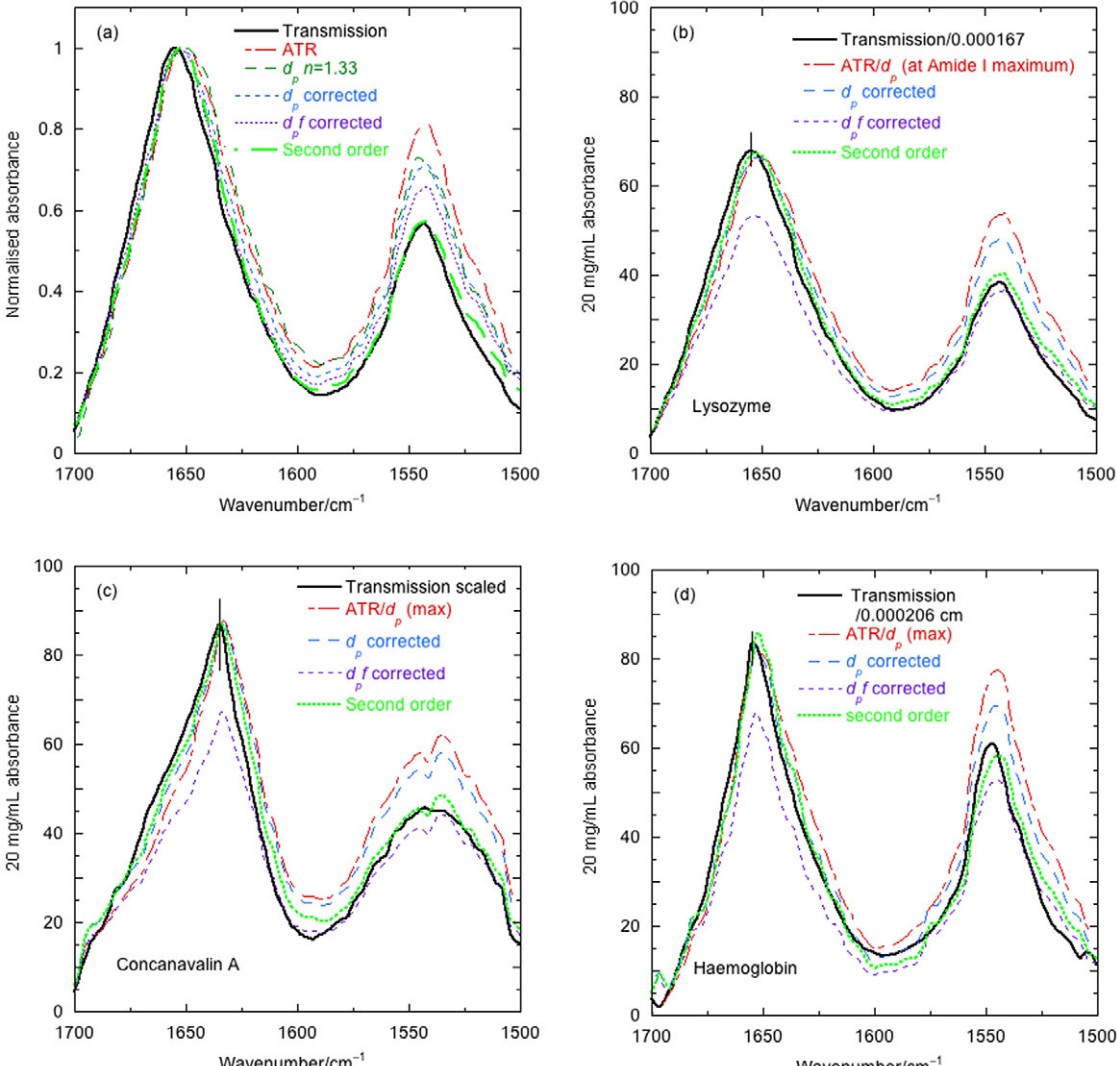

**Fig. 4.** Overlay of baseline-corrected 20 mg ml$^{-1}$ spectra with different corrections calculated as for Fig. 2. (*a*) Lysozyme normalised to 1, (*b*) lysozyme full scale, (*c*) concanavalin A full scale and (*d*) haemoglobin full scale. Transmission spectra are divided by a path length estimated from the water spectrum used for the baseline. Vertical black lines are an estimate of the pathlength and baseline correction error for the transmission spectra. ATR spectra are presented as data divided by $d_p$ at the protein absorbance maximum (~1,645 cm$^{-1}$) to facilitate comparison. This *arbitrary* choice leads to a magnitude match at the maximum. $d_p$ $n$ = 1.33 denotes $d_p$ calculated assuming the refractive index has no wavenumber dependence. $d_p$ denotes a calculation using literature wavenumber-dependent refractive index. $d_p f$ is first-order correction of Eq. (11).

correction to give an answer we would like to see, we simply note that quite small changes in water vapour and baseline move the precise position of peaks, especially for transmission spectra. In our experience, maxima of 18 independently collected lysozyme spectra ranged by 3 cm$^{-1}$ (data not shown). This emphasises the importance of having magnitude information retained for analysis.

## Discussion

The main conclusion from this work on the analysis of water transmission and ATR IR spectra is that we have presented equations that for the first time can be used to transform transmission to ATR IR absorbance spectra using Eq. (9). The second-order correction of Eq. (11) is a reasonable approximation for the converse ATR to transmission transformation. However, the transformations are only as good as the available refractive index data and literature data at absorbance maxima vary by a few percent. As a result of confidence in our ATR-IR transformations, a refined water

refractive index spectrum that is a hybrid of two literature spectra is available in the supplementary material. A constant value of the refractive index can only be assumed for samples with very small experimental absorbances.

Transformation of an ATR spectrum to transmission also depends on the angle of incidence of the light passing from air into the ATR crystal. By optimising the overlay of the transmission spectrum and the transformed ATR spectrum, we determined our angles of incidence on two systems to be 44.7° and 48°. This is an indirect benefit of this work, as in our experience, manufacturers simply declare the value is 45° and do not provide a method for determining it.

When we come to consider proteins in the amide I region, the absorbance and refractive indices are dominated by the contribution of water. It therefore follows that the refractive index is very close to that of water. We have shown that for the amide I band of aqueous protein samples (up to at least 50 mg ml$^{-1}$), Eq. (12) gives the transformation from ATR to transmission with

significantly less variation than we get from the buffer and water vapour corrections required for data analysis. Further improvements would follow by including protein absorbance into the refractive index.

If in practice, only the shape of the amide I band is required – this is current practice as all spectra are usually normalised to 1 – then, as long as wavenumber-dependent water refractive indices are used, one can get away with only the first-order correction (the $d_p \text{f}$) or even just $d_p$. However, if the information contained in the intensity is also important, or both amide I and amide II bands are to be considered, then the $d_p \text{f}$ and second-term corrections are required.

For future protein structure–fitting work using IR spectra, we therefore recommend the simplicity of ATR-IR data collection followed by transformation to transmission shape and intensity using Eq. (12). Eq. (12) in turn requires $d_p$ (Eq. (2)) and $\text{f} = \text{f}_{AtoC}(\text{f}_{xx} + \text{f}_{yy} + \text{f}_{zz})$ (Eqs. (5) and (6)), which depend on the refractive indices of water and the ATR crystal and the angle of incidence.

Finally, it is important to note that in contrast to transmission spectroscopy, in ATR spectroscopy, the evanescent field has different amplitude electric field components along $x$, $y$ and $z$. If one's sample is unoriented (or largely so) then that is not a problem, however, if one's sample is oriented, then it will interact to different extents with light polarised in the $x$, $y$ and $z$ directions, giving a spectrum that is a combination of absorption and linear dichroism. Given the penetration depth in our experiments is ~1,000 nm and a typical protein is 5–10 nm in size, we can ignore an oriented monolayer or two, even allowing for the higher intensity electric field close to the surface. However, if an oriented structure propagates, ATR is unlikely to be an appropriate technique for accurate spectroscopy. Germanium ATR crystals have higher refractive index than ZnSe, which is attractive for reducing the wavenumber dependence of $\text{f}$, but its smaller penetration depth means any surface effects are more significant.

**Open Peer Review.** To view the open peer review materials for this article, please visit http://doi.org/10.1017/qrd.2020.11.

**Supplementary Materials.** To view supplementary material for this article, please visit http://doi.org/10.1017/qrd.2020.11.

**Acknowledgements.** Helpful discussions with Marco Pinto Corujo, Don Praveen Amarasinghe and Dale Ang are gratefully acknowledged.

**Conflict of interest.** The authors have no conflicts of interest to declare.

**Author contributions.** ARo and ARe conceived the project. ARo and MS derived the theory and wrote the article. SG and NC designed experiments and collected data.

**Financial support.** Funding from the Engineering and Physical Sciences Research Council (grant EP/K007394/1) *via* the MOAC Doctoral Training Centre for DPA (EP/F500378/1) and via the Molecular Analytical Sciences Centre for Doctoral Training for MP (EP/L015307/1) and from the Biotechnology and Biological Sciences Research Council (BB/F011199/1) is gratefully acknowledged.

### Abbreviations

| | |
|---|---|
| ATR | attenuated total reflectance |
| IR | infrared |
| $i$ | incident |
| $j$ | $\sqrt{-1}$ |
| $t$ | transmitted |

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

## Appendix

This appendix presents a derivation of equations used in the main text. Although some of what follows may be found in the literature (e.g. Harrick, 1965; Milosevic, 2004), we have been unable to find their full derivations and found that the literature equations miss some factors needed for quantitative transformation rather than just ratios. In addition, assumptions about low absorbance are usually made, which are simply not valid for water. We here use the Maxwell boundary conditions on the electric and magnetic fields at the crystal surface boundary as the starting point.

The structure of what follows is first a discussion of absorbance in a linear transmission mode followed by ATR. Refractive index is then explored as a means of getting to the light intensity at each height above the ATR crystal in the presence of an absorbing sample. Finally, the ATR absorbance equations are derived to determine equations for what the spectrometer plots as absorbance (i.e. the log of the fraction of light that *does* reach the detector). The analysis could work for any solvent but is applied here to water using literature refractive index data. We use our analysis and experimental spectra to select literature values for the refractive index at different frequencies.

## Equations for absorbance

Absorbance occurs when the electric field of light interacts with a molecule causing it to jump to an excited state. Usually, the light–molecule coupling is dominated by the electric dipole transition moment, $\boldsymbol{\mu}$. Within a quantum electrodynamic formalism, the transition rate from $|0\rangle \rightarrow |f\rangle$ follows the Fermi golden rule (Craig and Thirunamachandran, 1984; Tokmakoff, 2009)

$$\Gamma = \frac{NB\,\mathtt{I}(\omega)}{3c},\tag{13}$$

where

$$B = \frac{1}{2\varepsilon_0\hbar^2}\left|\hat{e}\cdot\boldsymbol{\mu}^{f0}\right|^2\tag{14}$$

is an analogue of the unpolarised Einstein B-coefficient. Here, $N$ is the number of absorbers, $\varepsilon_0$ the permittivity of vacuum, $\hat{e}$ the unit vector parallel to the electric field of the light and $\boldsymbol{\mu}^{f0}$ the electric dipole transition moment from the ground state to the final state. The intensity of the light, $\mathtt{I}(\omega)$, is the time averaged value of the Poynting vector, $\mathbf{E}\times\mathbf{B}$, where $\mathbf{E}$ is the electric field vector and $\mathbf{B}$ the magnetic field vector and

$$\hat{\mathbf{k}}\times\mathbf{B} = \varepsilon_0 c_0 n\mathbf{E},\tag{15}$$

for $\hat{\mathbf{k}}$, the unit vector parallel to the direction of propagation. Thus,

$$\mathtt{I}(\omega) = \frac{\varepsilon_0 c_0}{2}n|\mathbf{E}(\omega)|^2.\tag{16}$$

So, to allow for complex fields and wavefunctions (though they are not needed here), we use tensor notation and write

$$\begin{aligned}B\mathtt{I}(\omega) &= \frac{1}{2\varepsilon_0\hbar^2}\boldsymbol{\mu}^{0f}\boldsymbol{\mu}^{f0}:\hat{e}\hat{e}^*\,\mathtt{I}(\omega)\\&= \frac{\varepsilon_0 c_0 n}{4\varepsilon_0\hbar^2}\boldsymbol{\mu}^{0f}\boldsymbol{\mu}^{f0}:\mathbf{E}\mathbf{E}^*,\end{aligned}\tag{17}$$

where inverting the order of the states of the transition dipole moments means taking the complex conjugate (denoted by *) of any complex wavefunctions.

## Transmission absorbance

When light passes through a sample, if it is the right frequency, it may be absorbed and its intensity reduced as it proceeds along the light path ($x$). In a transmission experiment, we usually ignore the polarisation information in Eq. (17) unless we need it and we replace $N$ in Eq. (13) with $C$ and write, for light propagating along the $x$-direction,

$$d\mathtt{I}_x(\omega) = -\kappa C\mathtt{I}_x(\omega)\,dx,\tag{18}$$

where $\kappa$, the Eulerian extinction coefficient, is a constant for a given sample and frequency (as long as molecules do not interact). The light at the exit edge of the sample, $\mathtt{I}_\ell$, is therefore related to the intensity at the entrance edge, $\mathtt{I}_0$, by the Beer–Lambert law

$$\ln\left(\mathtt{I}_0/\mathtt{I}_\ell(\omega)\right) = \kappa C\ell\tag{19}$$

for a sample of path length $\ell$. $\varepsilon$, the standard decadic extinction coefficients is related to $\kappa$ by

$$\kappa = \ln 10\varepsilon\tag{20}$$

since $\log_{10}(\mathtt{I}_0/\mathtt{I}_\ell) = \log_{10}e\ln(\mathtt{I}_0/\mathtt{I}_\ell)$ is what spectrometers output as absorbance, $A$. $\mathtt{I}_0$ at the entry edge of the sample is reduced by $(2/(1+n_t))^2$ (Eq. (39)) when it crosses the air/sample boundary at normal incidence. However, as the detector is outside the sample and the absorbance is linear with concentration, the inverse scaling factor is operative as the light emerges from the sample so, for transmission, we may ignore the intensity change at the interfaces of the sample and the air.

## ATR absorbance

In ATR spectroscopy, what we measure is more complex than in transmission: the light beam does not pass *through* the sample, rather the evanescent wave has $x$, $y$ and $z$ components of the electric field that interact with molecules in the sample leading to absorbance if frequencies match energy-level gaps. Further, the evanescent wave decays exponentially from the surface whether absorbance occurs or not. We use $i$ for light within the crystal and $t$ for light transmitted above the surface. Although we are interested in isotropic samples and unpolarised light in this work, the initial equations below can be applied to oriented samples. To derive Eqs. (9) and (12) of the main text, it is convenient to proceed in three stages:

(i) understand the electric field and penetration depth of the evanescent light in the absence of an absorbing sample,
(ii) express the evanescent light in terms of the incident (air) values and
(iii) determine what the instrument outputs as *nominal ATR absorbance*.

We note below where our equations are in accord with the published ones of Harrick (Harrick, 1960, 1965; Harrick and du Pré, 1966) and where they diverge for purposes of calculating absorbance intensity. Aspects of our work benefited from the following references (Kirkwood, 1937; Moffitt and Yang, 1956; Kirk and Klyne, 1974; Davidsson and Nordén, 1976; Barron, 1982; Craig and Thirunamachandran, 1984; Marsh *et al.*, 2000; Barth, 2007; Haris, 2013; Ogilvie and Fee, 2013).

## Refractive index as a function of wavenumber

For refractive index in the crystal, we write

$$n_i = c_0/c_i\tag{21}$$

and in the sample

$$n_t = c_0/c_t.\tag{22}$$

For this work, we used literature values for water refractive index (Querry, 1969; Querry *et al.*, 1969; Bertie and Eysel, 1985; Bertie *et al.*, 1989; Max and Chapados, 2009) and ZnSe (Madelung *et al.*, 1999) (which hardly changes) linearly interpolating between published data points (Fig. A1*a*). A further correction could be undertaken to improve $n_t$ for a protein sample by first estimating it, assuming that the aqueous protein refractive index is the same as that of water, then iteratively recalculating $n_t$ using a newly transformed ATR-to-transmission spectrum to include the protein absorbance using a Kramers–Kronig (Ohta and Ishida, 1988; Max and Chapados, 2009; Ogilvie and Fee, 2013) transformation,

$$n_t(\omega_0) = n_{t\infty} + \frac{\ln 10}{\pi} \wp \int_0^\infty \frac{\varepsilon(\omega)\, c C}{(\omega^2 - \omega_0^2)}\, d\omega, \qquad (23)$$

where $n_{t\infty}$ is the sample refractive index at infinite frequency and $\wp$ denotes the Cauchy Principal value. In practice, for protein concentrations less than 50 mg ml$^{-1}$, the difference is negligible (data not shown), so we worked with water values for our protein solutions.

Literature values for the refractive indices of water vary, especially in regions of high absorbance, so we began with the data from Bertie *et al.* and refined them by using the data of Max and Chapados at high wavenumber to give an improved transmission to ATR transformation for water. We confirmed the improvement by using it on data from a different instrument and ATR unit (Fig. A1*b,c*). The Fig. 2 and Fig. A1 plots were generated by determining the value of $\theta_i$ which minimised the difference between water ATR and transformed transmission spectra: PIKE ATR $\theta_i = 44.7°$ and Specac ATR $\theta_i = 48.0°$. The success in transforming the Specac ATR unit data using the hybrid water refractive index suggested by the Pike ATR unit data (purple dashed line of Fig. A1*c*) indicates that the error in the transmission to ATR transformation is less than 0.5%. The second-order (Eq. (10)) ATR to transmission transformation has a similar error. The experiment-to-experiment variation and water baseline-correction error of an ATR experiment is typically 1–3% depending on the operator.

## The electric field and penetration depth of the evanescent light in the absence of an absorbing sample on the ATR crystal

### The fields

Consider refraction at the crystal boundary (Fig. 1): the incoming wave has wave vector $\mathbf{k}_i$ with an angle of incidence $\theta_i$ to the normal. The interface is in the $x$–$y$ plane with the normal along $z$. The incoming electric and magnetic fields are, respectively,

$$\mathbf{E}_i(r,t) = \frac{\mathbf{E}_{i0}}{2}\left[ e^{\,j(\mathbf{k}_i \cdot \mathbf{r} - \omega t)} + c.c. \right] \qquad (24)$$

$$\mathbf{B}_i(r,t) = \frac{\mathbf{B}_{i0}}{2}\left[ e^{\,j(\mathbf{k}_i \cdot \mathbf{r} - \omega t)} + c.c. \right], \qquad (25)$$

where *c.c.* denotes complex conjugate, $\mathbf{r} = (x,y,z)$ is the position in space, $\mathbf{E}_{i0}$ is the electric field in the crystal incident on the crystal-sample boundary, $j = \sqrt{-1}$ and $k_0 = 2\pi/\lambda_0 = 2\pi v/c_0 = \omega/c_0$ is the magnitude of the wave vector of light in vacuum, *etc.*

Without loss of generality, we take the incident wave to propagate in the $x$–$z$ plane

$$\mathbf{k}_i = k_i \hat{\mathbf{k}}_i = (k_{ix}, 0, k_{iz}) \qquad (26)$$

and, similarly, above the surface but using the subscript *t*. Temporal translational invariance requires that the frequency (but not necessarily speed) of the light is unchanged on transmission across the crystal surface. Spatial translational invariance in the $x$–$y$ plane requires that the $x$-component of the wave vector is conserved

$$k_{tx} = k_t \sin\theta_t = k_i \sin\theta_i = k_{ix} = k_x, \qquad (27)$$

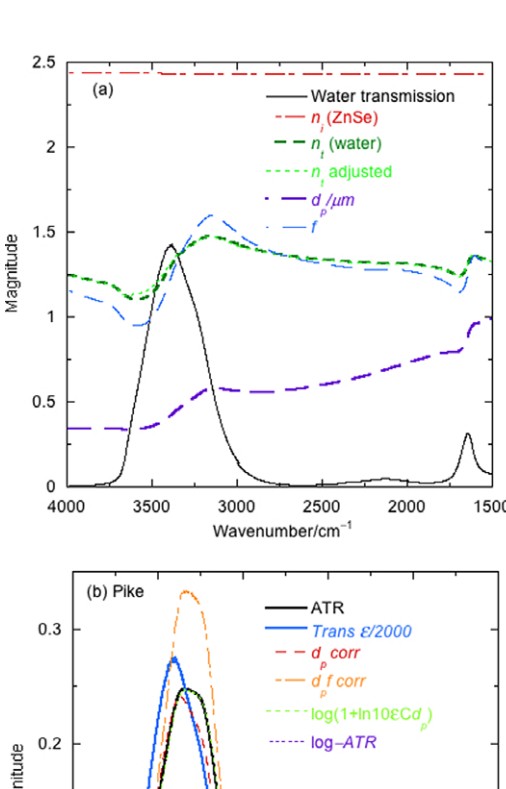

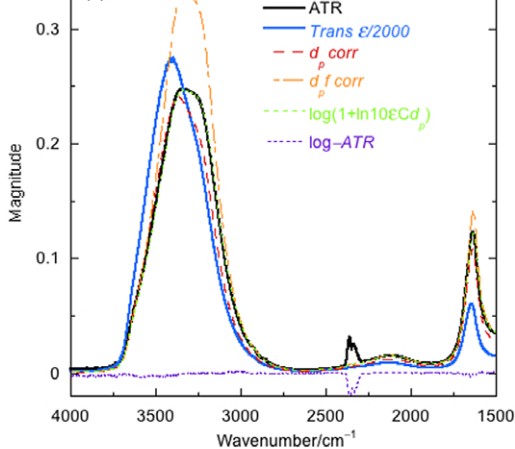

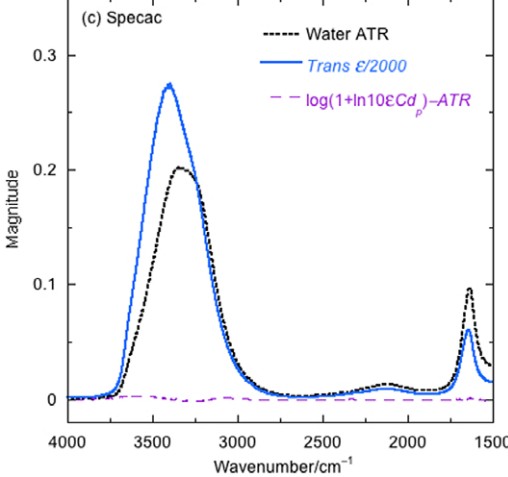

**Fig. A1.** (*a*) Wavenumber dependence of overlaid by water transmission absorbance; the refractive index of water derived from the data in Bertie *et al.* (1989) denoted Bertie and a combination of Bertie and Max and Chapados (Max and Chapados, 2009), denoted Bertie/Max; refractive index of ZnSe; $d_p$ and the unpolarised light intensity correction factor $f$ for water on our ATR unit. (*b*) and (*c*) Water ATR, transmission (scaled) and transmission converted to ATR using the combined Bertie/Max refractive index data for (*b*) a PIKE ATR single reflection unit in a Jasco V-470 and (*c*) a Specac Golden Gate single reflection unit in a Jasco FTIR-4200 (conversion spectrum not shown for clarity). Other parameters as for Fig. 2 in the main text.

which leads to Snell's law. We may also write

$$k_{tz} = \sqrt{k_t^2 - k_x^2} = k_i\sqrt{n_t^2/n_i^2 - \sin^2\theta_i}. \tag{28}$$

Above the critical angle, where total internal reflection occurs, it is better to write

$$k_{tz} = j\frac{2\pi}{\lambda_i}\sqrt{n_i^2\sin^2\theta_i - n_t^2} = j\alpha \tag{29}$$

where $\alpha$ is real and $j = \sqrt{-1}$. The analogue of Eq. (24) for the electric field above the crystal is thus

$$\begin{aligned}
\mathbf{E}_t(\mathbf{r},t) \\
= \frac{\mathbf{E}_{t0}}{2}\left[e^{\left[j(k_xx-\omega t)-k_i\sqrt{\sin^2\theta_i-n_t^2/n_i^2}\,z\right]} + e^{\left[-j(k_xx-\omega t)-k_i\sqrt{\sin^2\theta_i-n_t^2/n_i^2}\,z\right]}\right] \\
= \frac{\mathbf{E}_{t0}}{2}\left[\left(e^{\left[j(k_xx-\omega t)\right]} + e^{\left[-j(k_xx-\omega t)\right]}\right)e^{\left[-\alpha z\right]}\right].
\end{aligned} \tag{30}$$

So, with an ATR instrument, above the crystal surface, we have an evanescent wave which propagates along the $x$-axis, bound to the interface, decays exponentially in field strength with distance from the surface (Fig. 1b) and has nonzero field components in arbitrary directions, depending on the incident polarisation.

### The penetration depth

A common concept in ATR spectroscopy is the penetration depth, $d_p$ – the depth that enables the sample to 'see' the same flux of light as it would in a transmission experiment with pathlength $d_p$. It is the distance from the surface at which the light intensity has decayed to $1/e$ of its amplitude at the surface (see below).

In a transmission experiment, the total flux seen by a sample of path length $\ell$ is proportional to $\mathtt{I}_{t0}\ell/2$, with the factor of 2 coming from the time average of the wave (Schellman, 1975). By way of contrast, in ATR (assuming no absorbance), the electric field decays away from the surface with rate $-\alpha$ (Eq. (29)), so the light intensity (Eq. (16)) decays with rate $-2\alpha$. Writing $\mathtt{I}_{t0}$ as the intensity just on top of the crystal surface, the total energy flux seen by a (nonabsorbing) ATR sample is proportional to

$$\mathtt{I}_{t0}\int_0^\infty e^{-2az}\,dz = \mathtt{I}_{t0}/(2\alpha). \tag{31}$$

If we equate Eq. (31) with $\mathtt{I}_{t0}\ell/2$, it follows that $1/(\alpha)$ is an effective path length for low (strictly, non) absorbing samples. So, we write the penetration depth as

$$d_p = \frac{1}{\alpha} = \frac{1}{k_i\sqrt{\sin^2\theta_i - n_t^2/n_i^2}} = \frac{\lambda_0}{2\pi n_i\sqrt{\sin^2\theta_i - n_t^2/n_i^2}}. \tag{32}$$

The ATR electric field strength has decreased to $1/\sqrt{e}$ of its original value at this point and the light intensity to $1/e$ of its original value. This means that with $\theta_i = 45°$ for a ZnSe ATR crystal ($n_i = 2.4$) and $n_t \sim 1.3$ (refractive index of water away from absorbance (Max and Chapados, 2009; Kitadai *et al.*, 2014) and $d_p \sim 0.8\,\mu m$ (Fig. A1a). This is an ideal path length for infrared measurements on aqueous samples.

### Evanescent light in terms of the incident values

To calculate the nominal ATR absorbance (i.e. what the spectrometer plots), we need to know the light that is removed from the incident beam. We therefore need the electric field of the light that interacts with the sample. For $s$-polarisation ($E = E_o y$ in our axis system, Fig. 1) at the crystal surface ($z = 0$) from Maxwell's boundary conditions we write

$$E_{i0s} + E_{r0s} = E_{t0s}, \tag{33}$$

where $i$ is incident, $r$ is reflected and $t$ transmitted. The tangential magnetic field obeys a similar condition

$$B_{i0s}\cos\theta_i - B_{r0s}\cos\theta_i = B_{t0s}\cos\theta_t, \tag{34}$$

which, using Eq. (15), becomes

$$E_{i0s}n_i\cos\theta_i - E_{r0s}n_i\cos\theta_i = E_{t0s}n_t\cos\theta_t. \tag{35}$$

Similarly, for the perpendicular $p$-polarisation

$$E_{i0p}\cos\theta_i + E_{r0p}\cos\theta_i = E_{t0p}\cos\theta_t \tag{36}$$

$$B_{i0p} - B_{r0p} = B_{t0p} \tag{37}$$

and

$$E_{i0p}n_i - E_{r0p}n_i = E_{t0p}n_t. \tag{38}$$

These equations in different combinations lead to the Fresnel relations for transmitted (and reflected) waves including

$$t_s = \frac{E_{t0s}}{E_{i0s}} = \frac{2n_i\cos\theta_i}{n_i\cos\theta_i + n_t\cos\theta_t} \tag{39}$$

and

$$t_p = \frac{E_{t0p}}{E_{i0p}} = \frac{2n_i\cos\theta_i}{n_i\cos\theta_t + n_t\cos\theta_i}. \tag{40}$$

It follows that the amplitudes of the electric fields of the two polarisations at the surface, from Eqs. (30), (39) and (40), are

$$\begin{aligned}
E_{t0s} &= \frac{E_{i0s}n_i\cos\theta_i}{n_i\cos\theta_i + n_t\cos\theta_t} \\
&= \frac{E_{i0s}n_i\cos\theta_i}{\left(n_i\cos\theta_i + jn_t\sqrt{n_{it}^2\sin^2\theta_i - 1}\right)} \\
&= \frac{E_{i0s}n_i\cos\theta_i}{\left(n_i\cos\theta_i + j\sqrt{n_i^2\sin^2\theta_i - n_t^2}\right)}
\end{aligned} \tag{41}$$

$$\begin{aligned}
E_{t0p} &= \frac{E_{i0p}n_i\cos\theta_i}{n_i\cos\theta_t + n_t\cos\theta_i} \\
&= \frac{E_{i0p}n_i\cos\theta_i}{\left(jn_i\sqrt{n_i^2\sin^2\theta_i - 1} + n_t\cos\theta_i\right)} \\
&= \frac{E_{i0p}n_in_t\cos\theta_i}{\left(jn_i\sqrt{n_i^2\sin^2\theta_i - n_t^2} + n_t^2\cos\theta_i\right)}.
\end{aligned} \tag{42}$$

The $s$-polarised beam is pure $y$-directed, so

$$\mathbf{E}_{t0s} = \frac{E_{i0s}n_i\cos\theta_i\hat{\mathbf{y}}}{\left(n_i\cos\theta_i + j\sqrt{n_i^2\sin^2\theta_i - n_t^2}\right)}. \tag{43}$$

A $p$-polarised beam in the crystal has nonzero $x$- and $z$-components and below the surface is

$$\mathbf{E}_{i0p} = E_{i0p}(-\cos\theta_i\hat{\mathbf{x}} + \sin\theta_i\hat{\mathbf{z}}). \tag{44}$$

Above the surface, it follows that

$$\mathbf{E}_{t0p} = \frac{E_{i0p}n_in_t\cos\theta_i(-k_{tz}\cos\theta_i\hat{\mathbf{x}} + k_x\sin\theta_i\hat{\mathbf{z}})}{k_t\left(n_t^2\cos\theta_i + jn_i\sqrt{n_i^2\sin^2\theta_i - n_t^2}\right)}. \tag{45}$$

As

$$|k_t| = \frac{2\pi}{\lambda_t} = k_i \frac{\lambda_i}{\lambda_t} = k_i \frac{n_t}{n_i} \qquad (46)$$

$$\mathbf{E}_{t0p} = \frac{E_{i0p} n_i^2 \cos\theta_i (-k_{tz}\cos\theta_i \hat{\mathbf{x}} + k_x \sin\theta_i \hat{\mathbf{z}})}{k_i \left(n_t^2 \cos\theta_i + j n_i \sqrt{n_i^2 \sin^2\theta_i - n_t^2}\right)}. \qquad (47)$$

Thus for an unpolarised beam (equal intensity of *s* and *p*, giving a factor of $1/2$ in the component incident intensities), the corresponding light components are (*cf.* Eq. (16))

$$E_{t0x}E_{t0x}^* = \frac{2E_{i0p}^2 n_i^2 \cos^4\theta_i \left(n_i^2 \sin^2\theta_i - n_t^2\right)}{\left(n_i^4 \sin^2\theta_i - n_i^2 n_t^2 + n_t^4 \cos^2\theta_i\right)}$$

$$E_{t0y}E_{t0y}^* = \frac{2E_{i0s}^2 \cos^2\theta_i n_i^2}{\left(n_i^2 - n_t^2\right)} \qquad (48)$$

$$E_{t0z}E_{t0z}^* = \frac{2E_{i0p}^2 n_i^4 \cos^2\theta_i \sin^4\theta_i}{\left(n_i^4 \sin^2\theta_i - n_i^2 n_t^2 + n_t^4 \cos^2\theta_i\right)}.$$

It follows that since

$$E_{i0} = \frac{2E_{air}}{1 + n_i} \qquad (49)$$

(the air-crystal version of equation (39), at perpendicular incidence) following Eq. (16), we may write the energy density of the components as

$$U_{t0x} = \varepsilon_0 c_0 E_{i0}^2 \cos^2\theta_i n_i^2 n_t$$
$$\times \frac{\left(n_i^2 \sin^2\theta_i - n_t^2\right)}{\left(n_i^4 \sin^2\theta_i - n_i^2 n_t^2 + n_t^4 \cos^2\theta_i\right)}$$
$$= 2\mathtt{I}_{air}\left(\frac{2}{1+n_i}\right)^2 \cos^2\theta_i n_i^2 n_t \qquad (50)$$
$$\times \frac{\left(n_i^2 \sin^2\theta_i - n_t^2\right)}{\left(n_i^4 \sin^2\theta_i - n_i^2 n_t^2 + n_t^4 \cos^2\theta_i\right)}$$

$$U_{t0y} = \frac{\varepsilon_0 c_0 E_{i0}^2 \cos^2\theta_i n_i^2 n_t}{\left(n_i^2 - n_t^2\right)}$$
$$= 2\mathtt{I}_{air}\left(\frac{2}{1+n_i}\right)^2 \frac{\cos^2\theta_i n_i^2 n_t}{\left(n_i^2 - n_t^2\right)}$$
$$U_{t0z} = \varepsilon_0 c_0 E_{i0}^2 \cos^2\theta_i n_i^2 n_t \qquad (51)$$
$$\times \frac{n_t n_i^2 \sin^2\theta_i}{\left(n_i^4 \sin^2\theta_i - n_i^2 n_t^2 + n_t^4 \cos^2\theta_i\right)}$$
$$= 2\mathtt{I}_{air}\left(\frac{2}{1+n_i}\right)^2 \cos^2\theta_i n_t n_i^2$$
$$\times \frac{n_i^2 \sin^2\theta_i}{\left(n_i^4 \sin^2\theta_i - n_i^2 n_t^2 + n_t^4 \cos^2\theta_i\right)}$$

and

$$\mathtt{f}_x = \frac{2n_i^2 n_t \cos^4\theta_i \left(n_i^2 \sin^2\theta_i - n_t^2\right)}{\left(n_i^4 \sin^2\theta_i - n_i^2 n_t^2 + n_t^4 \cos^2\theta_i\right)}$$

$$\mathtt{f}_y = \frac{2n_i^2 n_t \cos^2\theta_i}{\left(n_i^2 - n_t^2\right)} \qquad (52)$$

$$\mathtt{f}_z = \frac{2n_i^4 n_t \cos^2\theta_i \sin^4\theta_i}{\left(n_i^4 \sin^2\theta_i - n_i^2 n_t^2 + n_t^4 \cos^2\theta_i\right)}$$

$$\mathtt{f}_{AtoC} = \left(\frac{2}{1+n_i}\right)^2. \qquad (53)$$

Combining these components, we may write the light intensity at the surface of the crystal to be

$$\mathtt{I}(z=0) = \mathtt{f}\,\mathtt{I}_{air}. \qquad (54)$$

where

$$\mathtt{f} = \left(\frac{2}{1+n_i}\right)^2 \cos^2\theta_i n_i^2 n_t$$
$$\times \left(\frac{\left(2n_i^2 \sin^2\theta_i - n_t^2\right)}{\left(n_i^4 \sin^2\theta_i - n_i^2 n_t^2 + n_t^4 \cos^2\theta_i\right)} + \frac{2}{\left(n_i^2 - n_t^2\right)}\right). \qquad (55)$$

## ATR nominal absorbance

At any point above the surface, the electric field has decayed exponentially and due to any absorbance. The absorbance decay rate is $2\kappa$, but it is modulated by the field intensity according to

$$\mathtt{I}(z) = \mathtt{I}(z=0)e^{-(2\alpha + 2\kappa C\mathtt{f})z}$$
$$= \mathtt{I}(z=0)e^{-(2\alpha + 2\varepsilon C\mathtt{f}\ln 10)z} \qquad (56)$$
$$= \mathtt{f}\,\mathtt{I}_{air}e^{-(2\alpha + 2\varepsilon C\mathtt{f}\ln 10)z}.$$

The light absorbed at each *z* for each wavenumber satisfies

$$d\mathtt{I}(z) = 2\kappa C\mathtt{I}(z)\,dz$$
$$= 2\kappa C\mathtt{f}\,\mathtt{I}_{air}e^{-(2\alpha + 2\kappa C\mathtt{f})z}\,dz. \qquad (57)$$

So, in the ATR experiment

$$\text{Total intensity absorbed} = \int_0^\infty 2\kappa C\mathtt{I}(z)\,dz$$
$$= -\mathtt{I}_{air}\frac{\kappa C\mathtt{f}}{(\alpha + \kappa C\mathtt{f})}e^{-(2\alpha + 2\kappa C\mathtt{f})z}\Big|_0^\infty$$
$$= \mathtt{I}_{air}\frac{\kappa C\mathtt{f}}{(\alpha + \kappa C\mathtt{f})} \qquad (58)$$
$$= \mathtt{I}_{air}\frac{\kappa C d_p \mathtt{f}}{(1 + \kappa C d_p \mathtt{f})}.$$

Thus, (assuming no air absorbance) what the spectrometer outputs as *Absorbance* when we do an ATR experiment is

$$A_{ATR} = \log_{10}\left(\frac{\mathtt{I}_{air}}{\mathtt{I}_{air}\left(1 - \frac{\kappa C d_p \mathtt{f}}{1 + \kappa C d_p \mathtt{f}}\right)}\right)$$
$$= -\log_{10}\left(1 - \frac{\kappa C d_p \mathtt{f}}{1 + \kappa C d_p \mathtt{f}}\right)$$
$$= -\log_{10}e\ln\left(1 - \frac{\ln 10\varepsilon C d_p \mathtt{f}}{1 + \ln 10\varepsilon C d_p \mathtt{f}}\right) \qquad (59)$$
$$= \log_{10}e\ln\left(1 + \ln 10\varepsilon C d_p \mathtt{f}\right)$$
$$\approx \varepsilon C d_p \mathtt{f} - \frac{\ln 10}{2}\left(\varepsilon C d_p \mathtt{f}\right)^2 + h.o.t.$$

with the final line being valid only for small $\varepsilon C d_p$. This looks comfortingly familiar apart from the factor $\mathtt{f}$ and the wavenumber dependence of $d_p$. Rearranging the final line of Eq. (59) gives

$$0 \approx \frac{\ln 10}{2}\left(\varepsilon C d_p \mathtt{f}\right)^2 - \varepsilon C d_p \mathtt{f} + A_{ATR}$$
$$\varepsilon C d_p \mathtt{f} \approx \frac{-1 \pm \sqrt{1 - 2A_{ATR}\ln 10}}{\ln 10} \qquad (60)$$
$$\varepsilon C \approx \frac{-1 \pm \sqrt{1 - 2A_{ATR}\ln 10}}{\ln 10\, d_p \mathtt{f}}.$$

So, the pathlength normalised transmission spectrum may be approximately determined from the nominal ATR absorbance signal using

 

$$\varepsilon C \approx \frac{1 \pm \sqrt{1 - 2A_{ATR} \ln 10}}{\ln 10 d_p \text{f}}$$

$$\approx \frac{A_{ATR}}{d_p \text{f}} + \frac{A_{ATR}^2 \ln 10}{2 d_p \text{f}} \tag{61}$$

(see main text). Therefore, assuming $n_t$ and thus $d_p$ are the same for both protein in buffer (*PW*) and buffer (*W*) and dropping the *ATR* subscript, the protein transmission spectrum may be determined from ATR data using

$$(\varepsilon C)_P = (\varepsilon C)_{PW} - (\varepsilon C)_W$$

$$\approx \frac{A_{PW} - A_W + A_{PW}^2 \ln 10/2 - A_W^2 \ln 10/2}{d_p \text{f}}$$

$$\approx \frac{A_P + (A_P + A_W)^2 \ln 10/2 - A_W^2 \ln 10/2}{d_p \text{f}} \tag{62}$$

$$\approx \frac{A_P}{d_p \text{f}} + \frac{A_P(A_P + 2A_W) \ln 10}{2 d_p \text{f}},$$

where $A_P$ is the baseline-corrected protein ATR spectrum and $A_W$ is the water (or appropriate buffer) ATR spectrum. Evaluating $d_p$ and f requires the refractive index of water. Fig. A1*a* shows this from Bertie's data at lower wavenumber and Max and Chapados' at higher wavenumber. The hybrid refractive index results in a transformed water transmission spectrum that better overlays the water ATR of Fig. 2*b* in the main text. Although we consider it to be more accurate, in practice as long as the same data are used for the reference set and sample spectra transformations any subsequent structure fitting error will be minimised.

## Area of the light beam

For completeness, we note that the light absorbed by a sample also depends on the area of the light beam. If the beam in the crystal has a circular cross-section of radius $R$, then the beam at the surface is an ellipse with short axis $D = 2R$ and long axis $H = 2R/\cos(\theta_i)$. So, the beam area increases with $\theta_i$. This parameter is the same for sample and baseline so can be ignored, but it does affect the signal to noise ratio.