## [Reviewer Report]

*Comments to Author*: The article derives equations for the transformation of ATR IR spectra of proteins to transmission spectra, allowing for more accurate quantitative structural measurements. The procedure derived by the authors corrects the standard procedure, widely accepted, of normalizing the intensities of proteins ATR spectra at the maximum absorption of amide I.

This article is written with great care and rigor. The authors correctly address the lack of coherent derivations in the literature for the ATR-to-transmission transformation. The work is inclusive and informative, and the modular structure of the work makes it so that it can be used both by experts and "novel" researchers in this field (the Appendix is essential in this sense). There is a didactic approach that many (even good) articles in the current scientific literature lack. However, I also have some considerations that the authors should address before the article can be published:

1)→Introduction: because a good part of the work is related to evaluating the contribution of water, I think that the Intro should provide a comparison of ATR vs Raman, as the latter is less influenced by the presence of water. In fact, Raman has been largely proposed for structural studies on proteins, see for instance Percot, A. et al. Water dependent structural changes of silk from Bombyx mori gland to fibre as evidenced by Raman and IR spectroscopies. Vibrational Spectroscopy 73, 79-89 (2014); Colomban, P., Dinh, H. M., Bunsell, A. & Mauchamp, B. Origin of the variability of the mechanical properties of silk fibres: 1 - The relationship between disorder, hydration and stress/strain behaviour. Journal of Raman Spectroscopy 43, 425-432 (2012); Yu, X. et al. Surface enhanced Raman spectroscopy distinguishes amyloid Β-protein isoforms and conformational states. Protein Science 27, 1427-1438 (2018).

2)→Discussion: the main concern I have here is as follows: the authors put great care and attention to correctly evaluate the optical phenomena that underpin the ATR-to-Transmission conversion, but they do not consider a major physical phenomenon related to ATR, i.e. pressure increase in the sample. The application of pressure during ATR experiments favors the formation of β-sheets (see for instance He, Z., Liu, Z., Zhou, X. & Huang, H. Low pressure-induced secondary structure transitions of regenerated silk fibroin in its wet film studied by time-resolved infrared spectroscopy. Proteins 86, 621-628 (2018)), and it has been shown that ATR spectra of proteins can overestimate some structures and underestimate others (see for instance Badillo-Sanchez, D., et al. Understanding the structural degradation of South American historical silk: A Focal Plane Array (FPA) FTIR and multivariate analysis. Scientific Reports 9, 17239-17249 (2019). This does not affect the accuracy of the optical corrections, but the final structural determination could be affected a priori by the pressure effects. The authors should discuss this point.

3)→Finally, even though the weight of errors (due to evaluation of RI and other entities) is somehow discussed, it would be beneficial to have clearer examples in the text of how such errors can impact quantitatively the final Transmission profiles.

---

## [Reviewer Report]

*Comments to Author*: I found this manuscript clearly written and what math I checked I verified was correct. The authors present a method for how to transform IR ATR measurement of aqueous solution of proteins into IR transmission spectra. They illustrate its use for three cases of interest: concanavalan A, hemoglobin, and lysozyme. The results obtained are very attractive, in my estimation. I do recommend publication but suggest that some small revisions will improve the presentation.

Here are my suggestions for revising the text:

1) name the proteins studied in the abstract

2) add to the abstract and the text more qualifications about how ATR might respond to alignment of the target analyte through a dependence on polarization. It is possible for the analyte to become aligned at the surface from several effects, such as surface adsorption, and the existence of a strong electric field caused by the formation of a double layer at the wall of the surface in contact with the aqueous protein solution. In this regard, any data that might be available on how salt concentration might change what is observed would be welcomed.

3) consider changing “aqueous proteins samples” in the caption to Fig. 1 to read instead “aqueous samples of three proteins”

---

## [Reviewer Report]

*Comments to Author*: Reviewer #1: The article derives equations for the transformation of ATR IR spectra of proteins to transmission spectra, allowing for more accurate quantitative structural measurements. The procedure derived by the authors corrects the standard procedure, widely accepted, of normalizing the intensities of proteins ATR spectra at the maximum absorption of amide I.

This article is written with great care and rigor. The authors correctly address the lack of coherent derivations in the literature for the ATR-to-transmission transformation. The work is inclusive and informative, and the modular structure of the work makes it so that it can be used both by experts and "novel" researchers in this field (the Appendix is essential in this sense). There is a didactic approach that many (even good) articles in the current scientific literature lack. However, I also have some considerations that the authors should address before the article can be published:

1)→Introduction: because a good part of the work is related to evaluating the contribution of water, I think that the Intro should provide a comparison of ATR vs Raman, as the latter is less influenced by the presence of water. In fact, Raman has been largely proposed for structural studies on proteins, see for instance Percot, A. et al. Water dependent structural changes of silk from Bombyx mori gland to fibre as evidenced by Raman and IR spectroscopies. Vibrational Spectroscopy 73, 79-89 (2014); Colomban, P., Dinh, H. M., Bunsell, A. & Mauchamp, B. Origin of the variability of the mechanical properties of silk fibres: 1 - The relationship between disorder, hydration and stress/strain behaviour. Journal of Raman Spectroscopy 43, 425-432 (2012); Yu, X. et al. Surface enhanced Raman spectroscopy distinguishes amyloid Β-protein isoforms and conformational states. Protein Science 27, 1427-1438 (2018).

2)→Discussion: the main concern I have here is as follows: the authors put great care and attention to correctly evaluate the optical phenomena that underpin the ATR-to-Transmission conversion, but they do not consider a major physical phenomenon related to ATR, i.e. pressure increase in the sample. The application of pressure during ATR experiments favors the formation of β-sheets (see for instance He, Z., Liu, Z., Zhou, X. & Huang, H. Low pressure-induced secondary structure transitions of regenerated silk fibroin in its wet film studied by time-resolved infrared spectroscopy. Proteins 86, 621-628 (2018)), and it has been shown that ATR spectra of proteins can overestimate some structures and underestimate others (see for instance Badillo-Sanchez, D., et al. Understanding the structural degradation of South American historical silk: A Focal Plane Array (FPA) FTIR and multivariate analysis. Scientific Reports 9, 17239-17249 (2019). This does not affect the accuracy of the optical corrections, but the final structural determination could be affected a priori by the pressure effects. The authors should discuss this point.

3)→Finally, even though the weight of errors (due to evaluation of RI and other entities) is somehow discussed, it would be beneficial to have clearer examples in the text of how such errors can impact quantitatively the final Transmission profiles.

Reviewer #2:

I found this manuscript clearly written and what math I checked I verified was correct. The authors present a method for how to transform IR ATR measurement of aqueous solution of proteins into IR transmission spectra. They illustrate its use for three cases of interest: concanavalan A, hemoglobin, and lysozyme. The results obtained are very attractive, in my estimation. I do recommend publication but suggest that some small revisions will improve the presentation.

Here are my suggestions for revising the text:

1) name the proteins studied in the abstract

2) add to the abstract and the text more qualifications about how ATR might respond to alignment of the target analyte through a dependence on polarization. It is possible for the analyte to become aligned at the surface from several effects, such as surface adsorption, and the existence of a strong electric field caused by the formation of a double layer at the wall of the surface in contact with the aqueous protein solution. In this regard, any data that might be available on how salt concentration might change what is observed would be welcomed.

3) consider changing “aqueous proteins samples” in the caption to Fig. 1 to read instead “aqueous samples of three proteins”